# Analyzing the Impact of Active Attack on the Performance of the AMCTD Protocol in Underwater Wireless Sensor Networks

**DOI:** 10.3390/s23063044

**Published:** 2023-03-11

**Authors:** Khalid Saeed, Wajeeha Khalil, Ahmad Sami Al-Shamayleh, Iftikhar Ahmad, Adnan Akhunzada, Salman Z. ALharethi, Abdullah Gani

**Affiliations:** 1Department of Computer Science, Shaheed Benazir Bhutto University, Sheringal Dir Upper 18000, Pakistan; khalidsaeed@sbbu.edu.pk; 2Department of Computer Science & Information Technology, University of Engineering and Technology, Peshawar 25000, Pakistan; wajeeha.khalil@uetpeshawar.edu.pk (W.K.); ia@uetpeshawar.edu.pk (I.A.); 3Department of Networks and Cybersecurity, Faculty of Information Technology, Al-Ahliyya Amman University, Amman 19328, Jordan; a.alshamayleh@ammanu.edu.jo; 4College of Computing & IT, University of Doha for Science and Technology, Doha P.O. Box 24449, Qatar; 5Department of Information System, College of Computers and Information Systems, Umm AL-Qura University, Al-lith 28434, Saudi Arabia; szharthi@uqu.edu.sa; 6Faculty of Computer Science and Information Technology, University of Malaya, Kuala Lumpur 50603, Malaysia; abdullah@um.edu.my

**Keywords:** security attack, active attack, routing attack, attacker nodes, malicious nodes

## Abstract

The exponentially growing concern of cyber-attacks on extremely dense underwater sensor networks (UWSNs) and the evolution of UWSNs digital threat landscape has brought novel research challenges and issues. Primarily, varied protocol evaluation under advanced persistent threats is now becoming indispensable yet very challenging. This research implements an active attack in the Adaptive Mobility of Courier Nodes in Threshold-optimized Depth-based Routing (AMCTD) protocol. A variety of attacker nodes were employed in diverse scenarios to thoroughly assess the performance of AMCTD protocol. The protocol was exhaustively evaluated both with and without active attacks with benchmark evaluation metrics such as end-to-end delay, throughput, transmission loss, number of active nodes and energy tax. The preliminary research findings show that active attack drastically lowers the AMCTD protocol’s performance (i.e., active attack reduces the number of active nodes by up to 10%, reduces throughput by up to 6%, increases transmission loss by 7%, raises energy tax by 25%, and increases end-to-end delay by 20%).

## 1. Introduction

Over the past ten years, interest in underwater wireless sensor networks (UWSNs) has grown significantly [1]. Due to the seas and water that cover it, the earth is known as the “blue planet” in the solar system [2]. Since the oceans cover more than 70% of our world [3,4], UWSNs are extremely important and an important research area [5]. Exploration underwater has increased significantly in the recent past [6]. UWSNs are made up of sensor nodes, and these nodes exchange data and communicate [7]. These sensor nodes have limited battery capacity [8]. In UWSNs, the sensor nodes also communicate with one another to determine the optimum path based on a set of criteria. The data are transmitted using the best route possible from the bottom of the sea to the surface level of the water, and then further [9,10,11,12]. UWSNs have the potential to explore hidden underwater resources. Additionally, they have enhanced capacity for both prediction and observation of the oceans [13]. Other significant UWSNs endeavors include ocean resource exploration, underwater information exchange, monitoring, and disaster avoidance [9,10,11,12,14,15]. Due to the constrained context of UWSNs, they are very different from conventional WSNs. Furthermore, compared to the context of WSNs, the issues faced by UWSNs are very different [16]. The major challenge in WSNs environment is the smaller battery power capacity of the nodes [17]. Some important research contributions in ad-hoc and sensor networks include improving fault tolerance [18], detection of faulty nodes in WSNs [19,20], reliable data transfer protocol for WSNs [21] and routing approach in IoT [22].

Due to a variety of factors, communication in the UWSNs context is difficult and unpredictable [23]. Due to absorption, high-level attenuation, and dispersion, radio waves in UWSNs are not effective. The atmosphere of UWSNs is suitable for the transfer of data via acoustic communication [24]. The range of frequencies that can be used for acoustic waves is limited, which affects their applicability in underwater communication [25]. In contrast to radio communication, where the speed of transmission is the same as the speed of light, the speed of propagation for acoustic communication is slower, i.e., five times slower than radio waves. The end-to-end and propagation delay in acoustic communication in UWSNs is greater due to the slower propagation speed of 1500 m/s. The usable bandwidth for audio communication is under 100 kHz. While most sensors in UWSNs are stationary, they can still move at a speed of 1 to 3 m/sec [16,26,27,28,29]. Figure 1 shows the UWSNs architecture. The sensor nodes deployed in UWSNs environment are labeled as sensor node A, B …. O and these nodes communicate under the water via acoustic waves.

Since the nodes utilized in UWSNs are operated by battery and rely on their remaining energy to function, maximum research in UWSNs focused on energy efficiency. Energy efficiency and security are equally important in the context of UWSNs. This research project, therefore, focuses on the effects of a security breach in a routing protocol in the context of UWSNs. The well-known routing system AMCTD [30] provides an effective routing solution for UWSNs environment. In order to investigate the effects of a security breach in a well-known routing system, this study actively attacked the AMCTD protocol. With the help of this work, the research community will better understand the effects of attacks in the UWSNs and will take required measures into account while building routing protocols for the UWSNs environment.

The rest of the research article is structured as follows: Related work is discussed in Section 2; Details regarding the AMCTD protocol are presented in Section 3; An active attack in the AMCTD protocol is discussed in Section 4; Performance evaluation and simulation parameters are mentioned in Section 5; Results are presented and discussed in Section 6; and the research is concluded in Section 7. Future research directions regarding this research are discussed in Section 8.

## 2. Related Work

Research conducted in [15] proposed an intrusion detection system (IDS) for reducing the bad influence of malicious nodes on the transmission of data. The mechanism of location monitoring is adopted in the proposed DOIDS. The DOIDS is suitable for underwater networks in which the deployment of sensor nodes is sparse. The malicious nodes are detected through the clustering algorithms known as DBSCAN. In order to reduce the DOIDS rate of false detection, a function is defined, known as the decision function. The obtained results show that the proposed algorithm significantly improved the accuracy rate of detection from 3% to 15% in different scenarios.

For the underwater acoustic sensor networks (UASNs) environment, the researchers in [31] developed a secure routing system. A brief signature scheme is provided for the purpose of establishing a secure path among the source and the destination nodes, because it is challenging to establish a trusted third party in UASNs. The researchers proposed a signature system that increases security and can fend against forgery attacks. A trusted third party online is not necessary for the suggested strategy. The authors provided a trapdoor approach to achieve anonymity among sensor nodes. The proposed routing protocol offers anonymity and mutual authentication between the origin and target nodes, preventing identity deception among sensor nodes. It also ensures security in the UASNs environment through the use of digital signatures and bilinear map trap-door technology. The overhead for managing a large number of pre-shared keys is decreased in the suggested method by the trap door. Opening the trap door includes one hash operation and one bilinear mapping. The NS2 simulator with UWSNs simulation package known as AquaSim has been used for the performance evaluation of the scheme proposed in this research. Throughput, consumption of energy and PDR are used to compare the performance of GPNC with LB-AGR. The outcomes demonstrate increased security and network performance for the suggested approach.

The authors in [32] proposed a distributed technique to defend against certain routing attacks in the context of UWSNs. The suggested mechanism can identify internal and external attacks on routing protocols, including wormhole and sinkhole attacks. The suggested technique uses two phases: silent surveillance and detection. For detection and mitigation, the sensor nodes eavesdrop on the communication of neighboring sensor nodes. Upon initial deployment, each sensor node identifies its neighbors using a secure protocol for discovering neighbors. The goal of neighbor activity surveillance is to find malicious UWSNs activity. As a result of the sinkhole attack, packets that are received may be altered or dropped. The method proposed by the research compares the incoming and outgoing traffic of each neighboring sensor node to identify sinkhole attacks. The signatures would not match if the malicious node lost or altered the packets, and an attack would be identified as a result. The mechanism suggested in this study can identify active attacks but not passive ones. For example, the proposed mechanism cannot identify a malicious node that captures traffic for analysis but does not interfere with it or drop it. By comparing the signatures, the suggested approach can also identify attacks that use encapsulated and out-of-bound wormholes. When a malicious node is found in the UWSNs environment, the network is isolated from the malicious node using an isolation strategy. As a result, the malicious node is prevented from taking part in UWSN activities and from interrupting routing operations. The Castalia simulator, which is based on OMNET++, was used to put the research theory into practice. This research can be expanded in the future by creating mechanisms for more attacks in the UWSN environment.

A protocol of secure discovery of neighbors was suggested in [33] for UASNs environment. The attacker can initiate a wormhole attack in a hostile environment if they learn that a nearby property is vulnerable. The consequences of the wormhole attack are undesirable outcomes that cannot be remedied by cryptographic methods. In this study, a class of protocols that perform secure neighbor discovery in UASNs and are resistant to wormholes were suggested. The protocols suggested by this research are based on the arrival-signal direction methodology. The suggested plan has the ability to withstand wormhole attacks. The four protocols that make up the suggested system are as follows: (i) In neighbor discovery, the following protocols must be used: (i) B-NDP requires two nodes; (ii) DV-NDP requires three nodes; (iii) SDV-NDP enhances DV-NDP; and (iv) MA-NDP, which allows for node mobility. The following are the evaluation findings for the four protocols: (i) B-NDP has an extremely high chance of being able to prevent fraudulent neighbors from establishing neighbor relationships. In B-NDP, the real neighbors can get to know one another. (ii) With a probability close to 1 and minimal links lost as a cost, DV-NDP has the potential to prevent fraudulent neighbors from establishing neighbor relationships. (iii) SDV-NDP has the potential to locate every wormhole; however, it loses many more links than DV-NDP. (iv) MA-NDP can control node mobility and has the ability to detect wormhole links that are randomly placed with high probability. B-NDP and MA-NDP protocols are appropriate for those applications whose main concerns are connectivity and end-to-end delay. Low density is also catered for with these protocols. DV-NDP and SDV-NDP protocols are appropriate for those applications that have wormhole resilience and high-level densities of nodes.

For the UASNs context, the authors of [34] recommended a security suite made up of both static as well as mobile nodes. The security suite’s goal is to protect UASNs environment confidentiality and integrity. Secure routing protocols and cryptographic basics are part of the security suite. First, the researchers suggested the FLOOD technique. A protocol is introduced with a secure variant called secure flood (SeFLOOD). The SeFLOOD protocol’s performance evaluation was carried out to determine how much overhead the FLOOD protocol required to become secure. The outcomes of the experiments show that the suggested suite is appropriate for the UASNs environment. The proposed suite has a lower level of communication overhead and requires less power consumption. The suggested suite needs less electricity and has lower communication costs. The suggested protocol suite’s main successes are listed below. (i) The proposed suite is effective due to the cypher text expansion’s modest impact. (ii) When compared to the unsecure protocol, the secure protocol’s discovery phase incurs 6% less overhead. (iii) The secure protocol’s reconfiguration phase produced no additional overhead when compared to the unsecure protocol. (iv) The secure protocol was created in accordance with Lampson’s advice on designing computer systems.

Researchers concentrated on DOS attacks in [35]. Flooding, man-in-the-middle, and demolition attacks are included in the classification of the DOS attack. Data that was exchanged between sensor nodes was captured by MITM attacks in UWSNs. In the context of UWSNs, wormhole, Sybil, and selective forwarding are all viable MITM attacks. By transmitting a continuous stream of packets to the base station during a flooding attack, the malicious node or nodes induce congestion. In the context of UWSNs, the flooding attack affects network performance across the board. In UWSNs, a demolishing attack entails changing or fiddling with the sensor node’s settings, which causes the network as a whole to be destroyed. A major factor in the demolition attack is physical security. In the UWSN context, mobile sensor nodes encounter problems like the out-of-coverage issue and the false neighbor identification issue. The authors of this study employed Aqua-Sim as a simulation tool. The findings show that the security method suited for mobile WSNs is not suitable for mobile UWSNs due to the performance differences between mobile WSNs and UWSNs. One potential direction for future research is to create secure UWSNs utilizing smart nodes and self-localization features to defend against denial of service (DoS) attacks in mobile UWSNs.

Research conducted in [36] proposed two algorithms such as multilayer sink (MuLSi) and MuLSi-Co. MuLSi-Co uses the cooperation technique, and it is the reliable version of MulSi. MuLSi proposes a network structure that is multilayered. The placement of the sink is optimum in order to avoid multi-hop communication. The nodes closest to the sink are considered as the best forwarder nodes. The scheme proposed in this research does not need node’s location information. The performance of network is better using MuLSi, but due to the single link, the operation is not reliable in MuLSi. Therefore, MuLSi-Co utilizes cooperation techniques in which the receiver has multiple data copies. Multiple copies of the same data are combined for the correct reception of data. The scheme proposed is better in terms of reliable data exchange and energy cost and has a smaller number of dead nodes. 

Research conducted in [37] presented better localization for UWSNs. The authors first presented algorithms of general localization, and then, they deployed beacon nodes for finding accuracy and error of the sensor localization. Then two more algorithms were presented, such as angle-based as well as distance-based localization algorithms. In this research, the realistic case is considered such as when the speed of sound in water is not known, and the sensor nodes are not synchronized. The simulation results reveal that the proposed algorithms are able to achieve better accuracy of localization.

The literature is summarized in Table 1.

## 3. AMCTD Protocol

The AMCTD protocol for UWSNs environments addresses the issues of short periods of stability, rapid energy consumption by nodes with low depth, and slow throughput during these unstable periods. By leveraging courier nodes to optimize movement under the network’s sparse conditions, AMCTD promotes global load balancing. The protocol enables efficient consumption of energy by the sensor nodes in the UWSNs environment. Figure 2a,b show the AMCTD flowchart and data transmission, respectively.

Hello packets are broadcasted by the sensor nodes in the first phase. Communication between sink, sensor and courier nodes starts up the network. Equation (1) is used by each sensor node to compute its weight.

Wi = (Priority value × Ri)/(Depth of water − Di)
(1)

where Wi represents weight of node i; Di represents depth of node i; Ri represents residual energy of node i; and priority value is constant.

Weight and depth of each sensor node are shared with the nearby sensor nodes during the setup phase. In this step, the neighbors within the transmission range are identified and placed in a separate queue in order to aid in finding the optimal forwarder. The sensor node transmits data towards the sink using the CSMA/CA technique in the next phase. The source node chooses the optimal forwarder by comparing their weights. The best forwarder node is the sensor node in the immediate vicinity with the highest weight.

Once the number of dead nodes in the UWSNs has increased by 2% during the weight update phase, each node then calculates its weight, utilizing Equation (2).

Wi = (priority value × Di)/Ri
(2)


The last step includes the motion of the courier sensor nodes, along with the variation in depth threshold of the nodes, designed to deal with the network sparsity.

Wi = Ri/(priority value × Di)
(3)


The AMCTD protocol in UWSNs is affected by noise, attenuation, path loss and signal to noise ratio.

## 4. AMCTD Protocol and Active Attacks

Various scenarios of active attacks with variations of attacker nodes were implemented in the AMCTD protocol. Scenarios include a combination of 4, 8 and 12 attacker nodes. Different combinations of attacker nodes were utilized for the performance assessment of AMCTD in different scenarios having different numbers of attacker nodes. In UWSNs, randomly chosen sensor nodes were set up as attacker nodes. These attacker nodes had more leftover energy than the other nodes in the UWSNs environment, which increased the likelihood that one of them would be chosen as a carrier node. If an attacker node is chosen as the carrier node, the data will not be sent to the sink node and will be dropped. The data will be transmitted to the sink if the real sensor node is chosen as a carrier node. In UWSNs environment, sensor node communication is multi-hop, meaning that if an attacker node is engaged during the data transmission among the source and sink node, the data will be lost. Performance will suffer if an attacker node is chosen as a carrier node. The AMCTD protocol will perform much worse the more attacker nodes it encounters. Figure 3 depicts the steps taken to implement an active attack using the AMCTD protocol.

## 5. Performance Evaluation and Implementation Parameters

The AMCTD protocol in UWSNs environments incorporated active attacks. Various types of attacker nodes were utilized in various scenarios to assess the effect of numerous attacker node variations on the AMCTD protocol in UWSNs environments. Table 2 reveals the simulation parameters utilized to perform simulations.

## 6. Results and Discussion

This section summarizes the research findings.

### 6.1. Evaluation of No. of Functioning/Alive Nodes

Figure 4 reveals the total no. of active nodes during the entire simulation. Findings indicate that there were 225 nodes at the beginning of the simulation. At the completion of the simulation in AMCTD, the results are as follows:85 alive/functioning nodes without attack,72 alive nodes with 4 attacker nodes,58 alive nodes with 8 active nodes,54 alive nodes with 12 active nodes.

These results demonstrate that active attack substantially influences energy usage, resulting in significantly fewer alive nodes in attack scenarios. It has been observed that by increasing the attacker nodes, significant increase has been observed in the number of dead nodes, and the number of alive/functioning nodes decreased substantially. When the number of alive/functioning nodes decreases, the overall network performance degrades.

**Figure 4 sensors-23-03044-f004:**
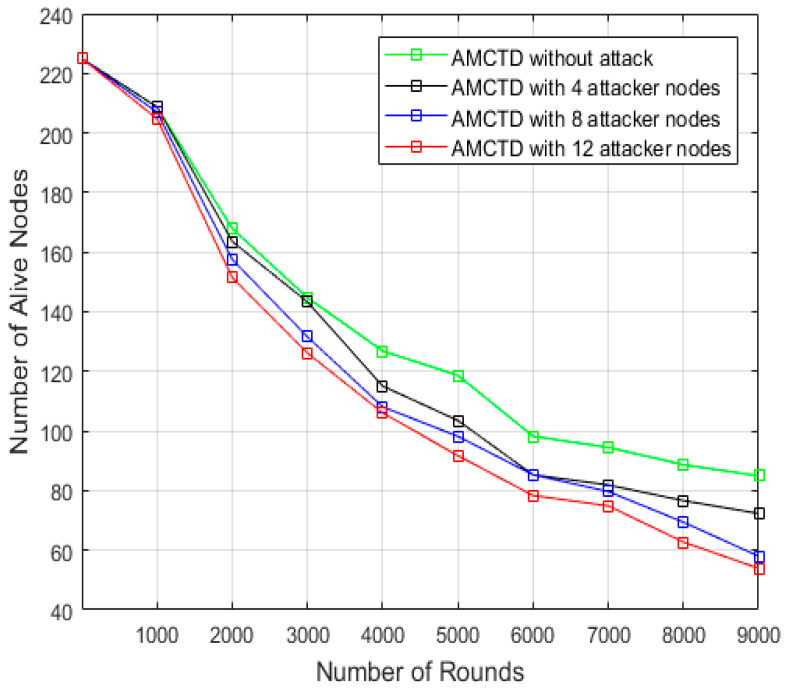
AMCTD: No. of Functioning/Alive Nodes Without attack and with 4, 8 and 12 Attacker Nodes.

The number of functioning/alive nodes in various scenarios is listed in Figure 5. The data represented in Figure 5 make it abundantly evident that as the no. of attacker nodes increases in the scenario, so does the no. of dead nodes. It indicates that in the scenarios, the no. of attacker nodes is inversely proportional to the no. of alive nodes and directly proportional to the number of dead nodes. Furthermore, the scenario’s increasing attacker nodes cause a 10% increase in dead nodes, which significantly reduces performance.

### 6.2. Analysis of Transmission Loss

The AMCTD protocol’s loss of transmission with different attacker nodes in various scenarios are explained in Figure 6. At the completion of the simulation in AMCTD, the results are as follows: It has much lower transmission loss without attack.It has increased transmission loss with 4, 8, and 12 attacker nodes.

This indicates the performance of the AMCTD protocol is reduced substantially in terms of transmission loss due to the active attack.

**Figure 6 sensors-23-03044-f006:**
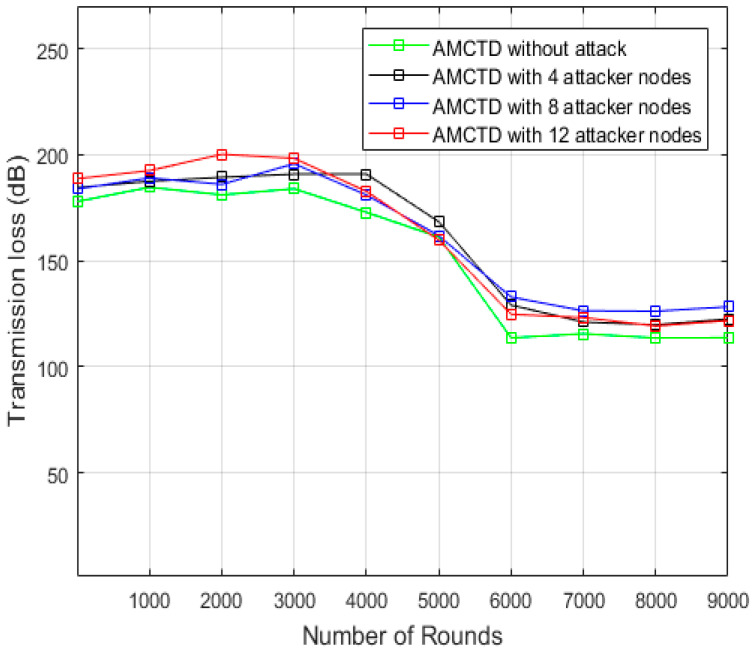
AMCTD: Transmission loss without attack and with 4, 8 and 12 Attacker Nodes.

Figure 7 displays the AMCTD protocol’s transmission loss under various conditions with various attacker node variations. The data in Figure 7 compares the without attacker scenario to active attack scenarios with 4, 8, and 12 attacker nodes; the transmission loss in the AMCTD scenario is much lower in withot attacker node scenario. Additionally, the transmission loss increases along with the number of attacker nodes in the scenario. Figure 7 makes clear that an active attack raises the AMCTD protocol’s transmission loss by up to 7%.

### 6.3. Analysis of Throughput

Different scenarios for the AMCTD protocol’s throughput, i.e., without attack, with 4, 8 and 12 attacker nodes, are plotted in Figure 8. According to the data, AMCTD with no attacker nodes has a higher throughput than AMCTD with 4, 8, or 12 attacker nodes. It demonstrates how the AMCTD protocol’s throughput performance suffers noticeably from active attack. Significant degradation in terms of throughput has been observed in the attack scenarios.

The throughput of the AMCTD protocol is described in Figure 9 for various circumstances. According to the findings in Figure 9, the AMCTD protocol’s throughput is 41.2% when there is no attack, 35.8% when there are 4 attacker nodes, 40.3% when there are 8 attacker nodes, and 37.8% when there are 12 attacker nodes. Furthermore, it is evident from Figure 9 that an active attack decreases the throughput of the AMCTD protocol by approximately 6%.

### 6.4. Analysis of Energy Tax

The AMCTD protocol was tested for the energy tax under four different scenarios, i.e., without and with 4, 8 and 12 attacker nodes. The energy tax value was revealed to be less for the without attacker node scenario as compared to all others having attacker nodes, in Figure 10. It is evident in Figure 10 that active attack significantly increases energy consumption.

Figure 11 displays the energy tax for the AMCTD protocol under various conditions. According to the findings in Figure 11, the AMCTD protocol has a much lower energy tax for 0, or without attack. Figure 11 makes it clear that an active attack raises the energy tax on the AMCTD protocol by up to 25%.

### 6.5. End-to-End Delay Analysis

End-to-end delay of AMCTD protocol is demonstrated in Figure 12 for variations of attacker nodes. The results depicted in Figure 12 demonstrate the significant influence that attacker nodes have on the end-to-end delay of the AMCTD protocol. In cases with four attacker nodes, eight attacker nodes, and twelve attacker nodes, there is a significantly increased end-to-end delay.

Figure 13 displays the AMCTD protocol’s end-to-end delay under various conditions, including AMCTD without an attack, 4, 8 and 12 attacker nodes. According to the findings in Figure 13 for AMCTD, the end-to-end delay without attack is 20% lower than it is in the 12 attacker nodes scenario. The findings in Figure 13 demonstrate that an active attack has a significant impact on the AMCTD protocol’s end-to-end delay.

## 7. Conclusions

The AMCTD protocol was evaluated under various conditions, including the presence and absence of attacker nodes. The performance of the protocol was evaluated in different scenarios involving attacker nodes, and it was found that active attacks significantly decreased the protocol’s performance across multiple parameters. The study found that active attacks resulted in

a reduction of up to 10% in the number of functioning nodes in the UWSNs environment;a decrease of up to 6% in throughput;an increase of up to 7% in transmission loss;a rise of up to 25% in energy cost; andan increase of up to 20% in end-to-end latency.

## 8. Future Directions and Challenges

We intend to offer safe routing solutions that are energy-efficient in the future for various environments. The routing solutions will include a protection mechanism to fend off various security intrusions. To ensure that the suggested solution is still appropriate for the UWSNs environment, the computation cost must be taken into account while creating a security mechanism. When designing a security solution for the UWSNs environment, energy efficiency and security should be balanced and should be treated as equally important.

Further exploration in this research area can be proposals for a key distribution scheme having support for different mobility models in the UWSNs environment, proposing more energy-efficient secure solutions for UWSNs. The following are the key future research issues:Secure and energy efficient solutions: to design secure UWSNs with intelligent sensor nodes and self-localization for combating DOS attacks in mobile UWSNs [35,38].Trust: to establish the trust when the nodes are moving in the underwater environment, establishment of trust when the sensor nodes are sparsely deployed and they are far away from each other, considering block cipher algorithms such as ARIR and SEED for UWSNs environment, using the technology for underwater security with other network systems such as IEEE 802.15.3 (UWB), IEEE 802.11 (WLAN), IEEE 802.15.4 (ZigBee) [39,40,41,42,43].Intelligent sensor environments: to use AI models for reducing intelligent attacks in the network leading to robust systems. The transfer rate of packets in UWSNs environment can be reduced by utilizing intelligent sensor nodes that are self-localized; to address the DoS problem in UWSNs environment, secure UWSNs having intelligent sensor nodes and self-localization should be designed [35].

## Figures and Tables

**Figure 1 sensors-23-03044-f001:**
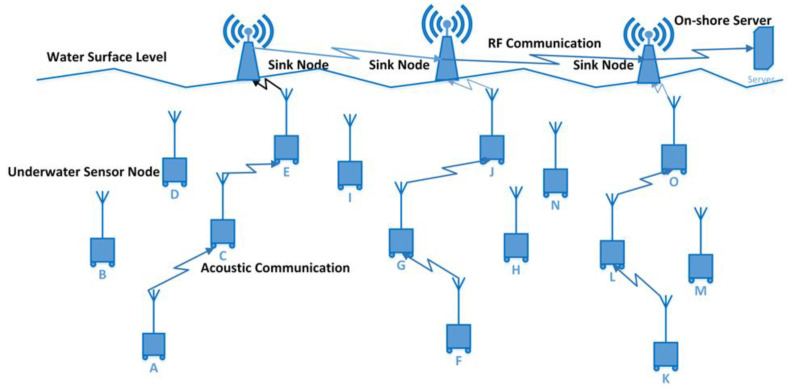
UWSNs Architecture.

**Figure 2 sensors-23-03044-f002:**
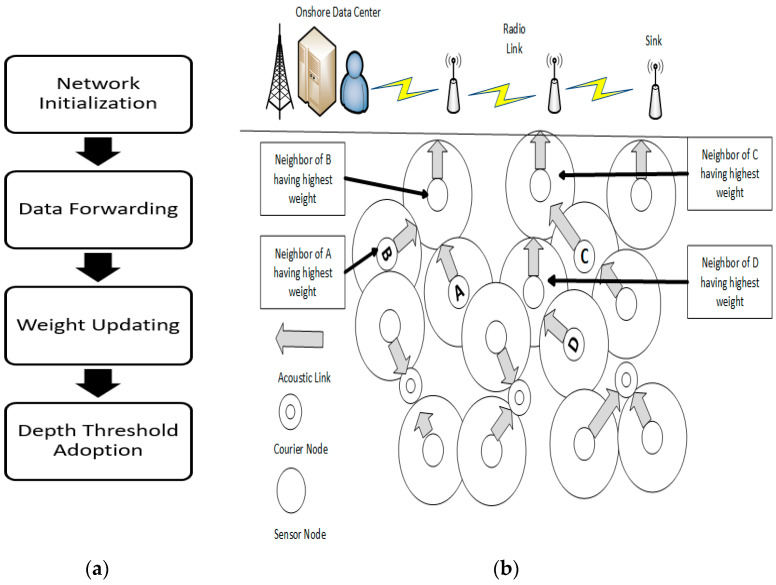
(**a**) Block diagram of AMCTD. (**b**) Data Transmission in AMCTD Protocol.

**Figure 3 sensors-23-03044-f003:**
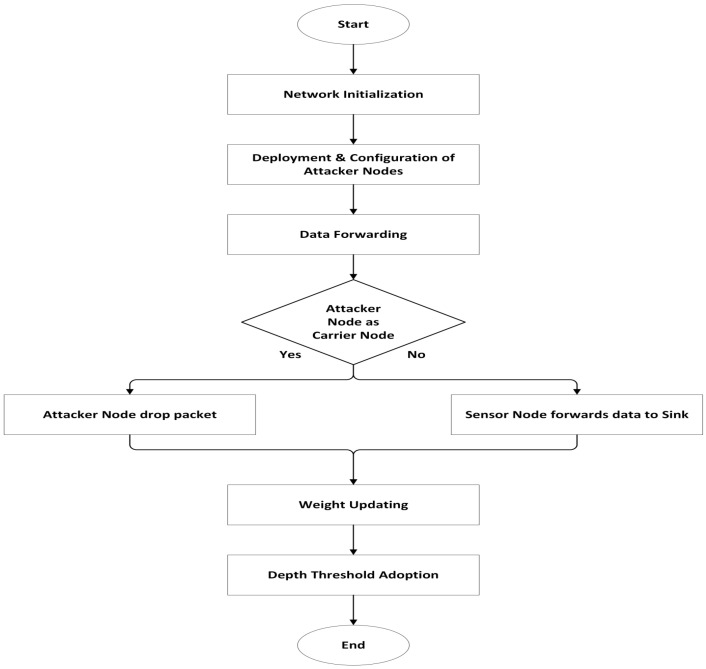
Active Attack in AMCTD.

**Figure 5 sensors-23-03044-f005:**
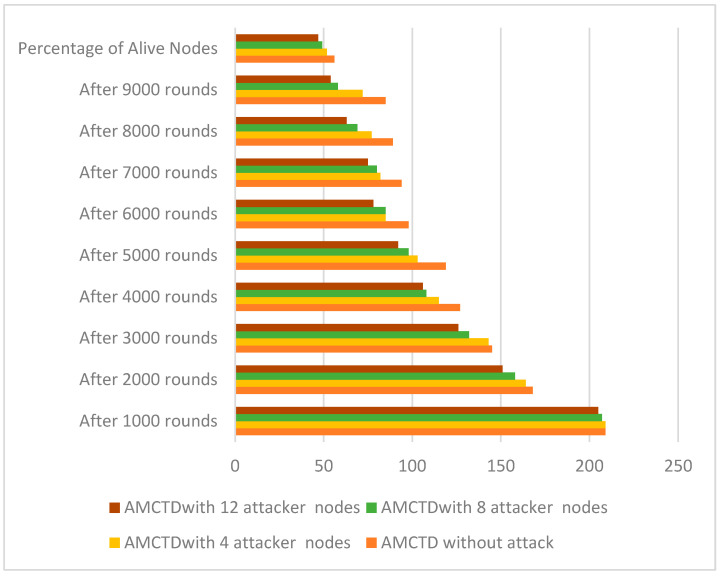
No. of Alive/Functional Nodes of AMCTD without Attack, AMCTD with 4, 8 and 12 Attacker nodes.

**Figure 7 sensors-23-03044-f007:**
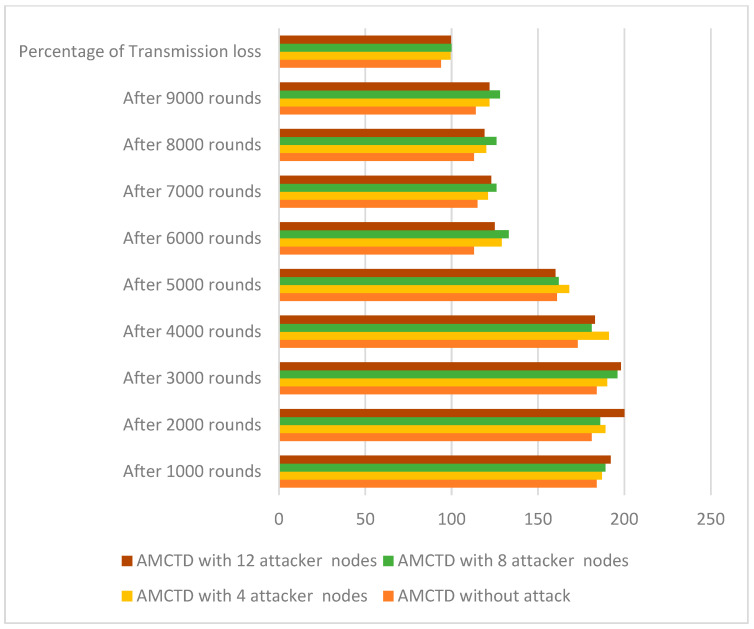
Transmission loss of AMCTD without Attack, AMCTD with 4, 8 and 12 Attacker Nodes.

**Figure 8 sensors-23-03044-f008:**
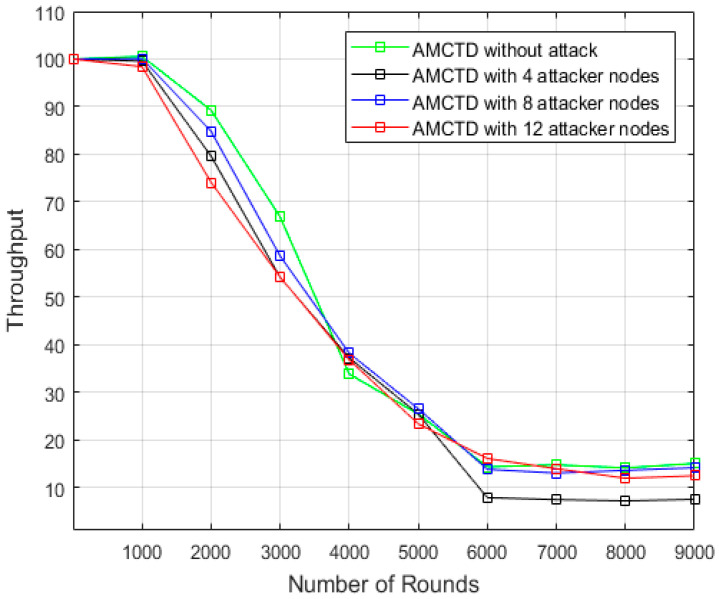
AMCTD: Throughput Analysis without attack and with 4, 8 and 12 Attacker Nodes.

**Figure 9 sensors-23-03044-f009:**
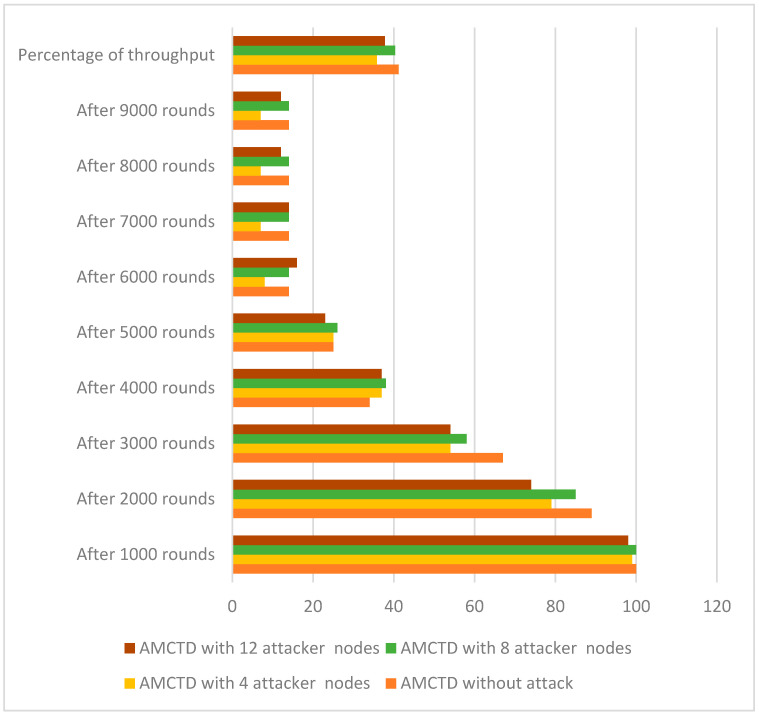
Throughput of AMCTD without Attack, AMCTD with 4, 8 and 12 Attacker Nodes.

**Figure 10 sensors-23-03044-f010:**
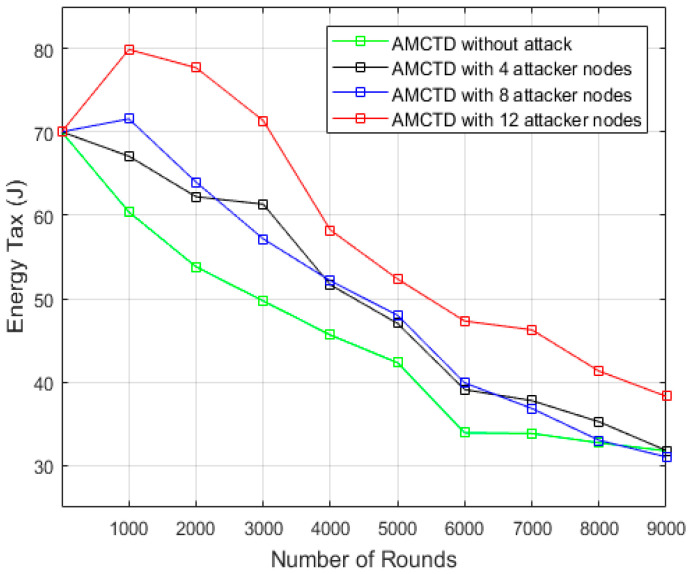
AMCTD: Energy Tax without attack and with 4, 8 and 12 Attacker Nodes.

**Figure 11 sensors-23-03044-f011:**
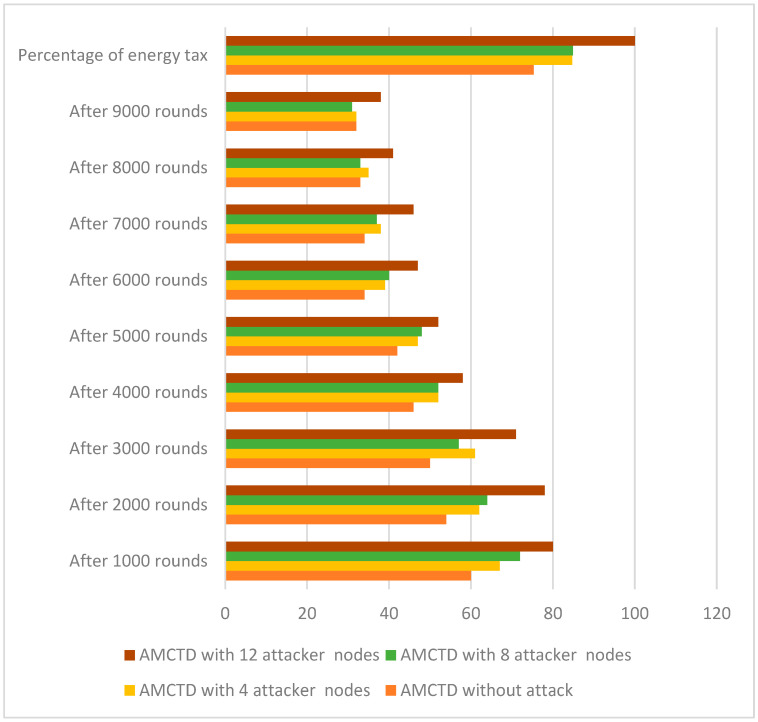
Energy Tax of AMCTD without Attack, AMCTD with 4, 8 and 12 Attacker Nodes.

**Figure 12 sensors-23-03044-f012:**
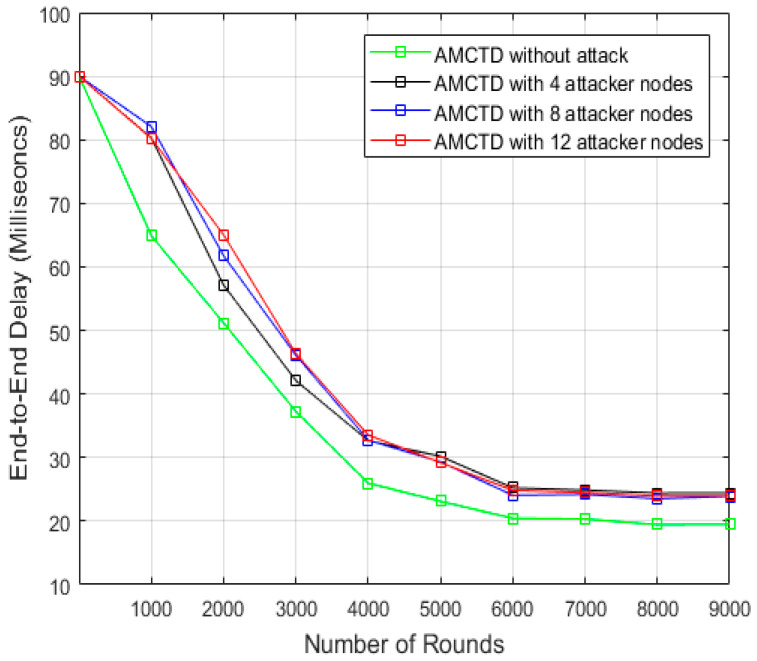
AMCTD: End-to-End Delay without attack and 4, 8 and 12 Attacker Nodes.

**Figure 13 sensors-23-03044-f013:**
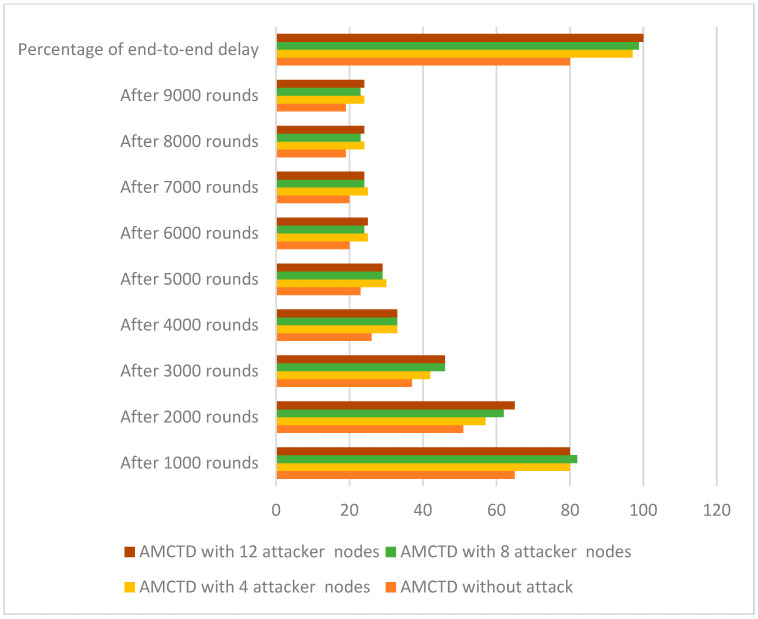
End-to-End Delay of AMCTD without Attack, AMCTD with 4, 8 and 12 Attacker Nodes.

**Table 1 sensors-23-03044-t001:** Summary of related work.

Technique	Contribution	Tool Used
IDS for Opportunistic Routing in UWSNs [15]	Proposed an Intrusion Detection System (IDS) for reducing the bad influence of malicious nodes on the transmission of data. The mechanism of location monitoring is adopted in the proposed DOIDS. The malicious nodes are detected through the clustering algorithms DBSCAN. The obtained results show that proposed algorithm significantly improved the accuracy rate of detection from 3% to 15% in different scenarios.	Not mentioned
Secure routing scheme for UASNs [31]	Recommended secure routing for UASNs. Signature algorithm is proposed for authentication between source and destination node. A trap-door scheme is used in order to achieve anonymity of the nodes.	NS2 with AquaSim
Securing network from routing attacks [32]	Proposed distributed approach for detecting and mitigating the routing attacks in UWSNs. An analytical model is proposed for the said purpose.	Castalia simulator
Secure discovery of neighbor in UASNs [33]	Proposed protocols suite for secure neighbor discovery in UASNs. The proposed protocols are based on the Direction of Arrival (DoA) signals approach.	C++
Secure communication suite for UASNs [34]	Proposed scheme includes secure routing protocol and cryptographic primitives. Proposed protocols suite has limited power consumption and overhead; that is why it is suitable for UASNs.	Real data used
Secure communication in mobile UWSNs [35]	Flooding attack in UWSNs is simulated, and its impact is analyzed on the performance of UWSNs. It has been concluded that techniques suitable for the WSN environment are not suitable for UWSNs environment.	Aqua-Sim
MuLSi-Co routing technique for UASNs [36]	Proposed two algorithms: multilayer sink (MuLSi) and MuLSi-Co. MuLSi-Co uses cooperation technique, and it is the reliable version of MulSi. The schemes proposed are better in terms of reliable data exchange and energy cost and have a smaller number of dead nodes.	MATLAB
Presented better localization for UWSNs [37]	The authors first presented algorithms of general localization. Then two more algorithms were presented: angle-based and distance-based localization algorithms. The simulation results reveal that the proposed algorithms are able to achieve better accuracy of localization.	Not mentioned

**Table 2 sensors-23-03044-t002:** Simulation Parameters.

Parameter	Value
No. of Nodes	225
No. of Sinks	10
Routing Protocol	AMCTD
Attack type	Active Attack
No. of Attacker nodes	4, 8, 12
No. of rounds	9000
Simulation volume	500 m × 500 m × 500 m

## Data Availability

Not applicable.

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
