# Peer review of "Analyzing the Impact of Active Attack on the Performance of the AMCTD Protocol in Underwater Wireless Sensor Networks"

_sensors, 2023, doi:10.3390/s23063044_

Round 1

Reviewer 1 Report

This paper presented an experimental approach to study the effect of active attacks on the cyber security of underwater sensor networks. The attack was implemented in the Threshold-optimized Depth-based Routing (AMCTD) protocol. The communication was exhaustively evaluated with/without an active attack with benchmark evaluation metrics. Overall, this paper has a smooth logic flow and provides technical details for the implementation and experimental study. It is recommended for the authors to consider highlight the scientific/technical contribution of this work compared to the literature and further comment on the boundary conditions for achieving the similar results. For instance, is there any limitation on the number of nodes? There are also a few minor typos to be corrected through the write-up, e.g. attach -> attack. 

Detailed comments:

1. It is suggested to convert Tables 2, 3, 4, 6 into bar graphs to improve the readability of the results.

2. Is it possible to include Fig. 2 and 3 in the same figure (as an and b) or put them side by side so that they can be easily compared with each other? 

Author Response

Review response forms have been prepared. We are thankful for the valuable suggestions by the reviewers. Our research group has tried our best to meet the expectations of the reviewers. The changes are marked with red font in revised manuscript.

Once again thank you for your support and encouragement.

Author Response

(The authors gave the same response as above.)

Reviewer 3 Report

I think this manuscript is well organized and well written. In the review process I have following comments.

1.      At the end of the introduction section, compare the  Related Work in a table.

2.      In Figure 2, the title of the flowchart should be changed to block diagram and the start and end sections should be removed.

3.      In Equation (1), if the value of Priority value is constant, what is the necessity of this parameter in the Equation?

4.      The quality of some figures is low and should be redesigned.(such as Figure 2 and Figure 3)

5.      Some references are out-of-date, so these references before 2000 should be deleted. At the same time, many important recent references are missing, which can support the idea of this paper, the following references should be added in the Section "Related Work":

"RDTP: reliable data transport protocol in wireless sensor networks", Telecommunication systems, 2015, DOI: 10.1007/s11235-015-0098-2

Improving fault tolerance in ad-hoc networks by using residue number system. Journal of Applied Sciences, 8(18), 3273-3278.

Improvement of fault detection in wireless sensor networks. In 2009 ISECS International Colloquium on Computing, Communication, Control, and Management (Vol. 4, pp. 644-646). IEEE.

A novel approach for faulty node detection with the aid of fuzzy theory and majority voting in wireless sensor networks. International Journal of Advanced Smart Sensor Network Systems, 2(4), 1-10.

An overlapping routing approach for sending data from things to the cloud inspired by fog technology in the large-scale IoT ecosystem. Wireless Networks, 28(2), 521-538.

Author Response

(The authors gave the same response as above.)

Round 2

Reviewer 2 Report

No more comments from my side.